# The Downregulation of Opioid Receptors and Neuropathic Pain

**DOI:** 10.3390/ijms24065981

**Published:** 2023-03-22

**Authors:** Lin Li, Jing Chen, Yun-Qing Li

**Affiliations:** 1Institute of Medical Research, Northwestern Polytechnical University, Xi’an 710072, China; 2Department of Anatomy, Histology and Embryology and K. K. Leung Brain Research Centre, The Fourth Military Medical University, No. 169, West Changle Road, Xi’an 710032, China

**Keywords:** neuropathic pain, opioids, opioid receptor, tolerance, downregulation, internalization

## Abstract

Neuropathic pain (NP) refers to pain caused by primary or secondary damage or dysfunction of the peripheral or central nervous system, which seriously affects the physical and mental health of 7–10% of the general population. The etiology and pathogenesis of NP are complex; as such, NP has been a hot topic in clinical medicine and basic research for a long time, with researchers aiming to find a cure by studying it. Opioids are the most commonly used painkillers in clinical practice but are regarded as third-line drugs for NP in various guidelines due to the low efficacy caused by the imbalance of opioid receptor internalization and their possible side effects. Therefore, this literature review aims to evaluate the role of the downregulation of opioid receptors in the development of NP from the perspective of dorsal root ganglion, spinal cord, and supraspinal regions. We also discuss the reasons for the poor efficacy of opioids, given the commonness of opioid tolerance caused by NP and/or repeated opioid treatments, an angle that has received little attention to date; in-depth understanding might provide a new method for the treatment of NP.

## 1. Introduction

Neuropathic pain (NP), which is defined by the International Association for the Study of Pain (IASP) as “pain caused by a lesion or disease of the somatosensory system”, represents a wide range of pain syndromes manifesting as a large number of symptoms and signs [1,2]. A lack of reliable epidemiological data on NP, especially about the prevalence and incidence rate among the general population, has plagued the field for years. Two large-scale population-based postal surveys conducted in Britain and France and a systematic retrospective study showed that the overall prevalence of NP could be as high as 7% to 10% in the general population [3,4,5]. Additionally, the quality of life of NP patients is significantly worse than that of individuals without NP (i.e., individuals without damaged or irritated nerves) due to the commonality and seriousness of sleep disturbances, anxiety, and depression, which also contribute a heavy economic and psychological burden [6,7,8,9].

According to the definition provided by the IASP, the typical signs and symptoms associated with the presence of NP include allodynia, hyperalgesia, and paresthesia [1,2]. NP reflects different pathological phenomena occurring within the nervous system and is usually described based on its different anatomical localizations and/or etiologies [10,11] as either peripheral (e.g., nerve, plexus, and nerve root) or central (spinal cord and brain), including metabolic imbalance, viral infection-related neuropathies, autoimmune disorders, chemotherapy-caused peripheral neuropathies, post-traumatic neuropathy, inflammatory disorders, inherited channelopathies, etc. The mechanisms underlying these different conditions are multiple; some are known, while many are not [10]. As a syndrome with multiple etiologies and complex etiopathogenesis, it is widely recognized that it is more effective to explore ways to alleviate and even cure NP by studying its pathogenesis [6,10,11]. Although years of continuous research have revealed a variety of possible pathogeneses for the development of NP and proposed many treatment methods, most people have ignored the importance of the opioid system in NP due to the low therapeutic efficacy, complex side effects, and high analgesic tolerance of opioids [10,11,12,13,14].

Tolerance refers to the reduction in or loss of drug efficacy caused by long-term administration and indicated by the pharmacological changes to the right of the dose-response curve, the development and degree of which depend on the drug interactions with the opioid receptors, drug dose, and frequency of administration [15]. Opioid tolerance is the most common analgesic tolerance, and the USA FDA has defined opioid tolerance as 60 mg morphine-equivalent daily; prescriptions that exceed 100 mg are subject to scrutiny for opioid abuse [16]. Many mechanisms contribute to opioid tolerance, including the enhancement of drug metabolism, compensatory/opponent processes, and the downregulation of opioid receptors [17]; the latter has long received widespread attention. Receptor downregulation has been defined experimentally as the decrease in the number of radioligand binding sites detected on the cytomembrane, which is generally caused by repeated and/or continuous activation of receptors [18]. In addition, there is also evidence that downregulation can be mediated by conformational changes (e.g., GRK-mediated phosphorylation) or internalization of receptors, which lead to the reduction in the sensitivity or activity of receptors without the loss of the total number of receptors [19,20,21]. In this review, we briefly cite some studies related to opioid receptors concerning NP to highlight the role of opioid receptors and to explain the poor therapeutic effect of opioids.

## 2. The Opioid System and Opioids in NP

### 2.1. Endogenous Opioid System

The endogenous opioid system is a highly diverse peptide neurotransmitter system comprising opioid receptors and three major neuropeptide families containing multiple member peptides; typically, β endorphins are derived from pro-opiomelanocortin, enkephalins are derived from proenkephalin and prodynorphin precursors, and dynorphins are derived from prodynorphin [22]. As another important component of the opioid system, there are three main canonical opioid receptors, namely μ, δ, and κ opioid receptors (MOR, DOR, and KOR, respectively), which play important roles in various physiological and pathological functions of the body (Table 1), as well as the nociceptin/orphanin FQ receptor (NOR), which was previously known as opioid receptor-like-1, and are widely distributed in systemic cells [12,13]. The endogenous opioid system has been studied for decades, as it plays an important role in analgesia, in addition to participating in learning and memory, immunological response, gastrointestinal function, and many other functions [23,24]. Opioid receptors are a class of classical G-protein-coupled receptors and are among the most widely expressed receptors in the nervous systems; the composition and distribution of different opioid receptor types vary depending on the region [25]. MOR was found to be expressed throughout the pain axis, the stimulation of which in both peripheral nociceptors and supraspinal structures can alleviate the nociceptive components [23,25]. DOR is highly expressed in the forebrain regions and regulates analgesia, predominantly under chronic pain conditions [26,27]. KOR is the same as MOR, which is heavily involved in the integration of reinforcement and aversion stimuli including severe and acute pain, and is mainly distributed in the brainstem, midbrain, and forebrain [28,29,30]. Although the pair of nociceptin and NOR was discovered last [31] and is mainly expressed in the spinal cord, brainstem, and forebrain [32,33], an increasing number of studies have indicated a dual role in pain and emotion [33,34,35], considered a possible molecular target for substance abuse treatment [36,37].

Opioid receptors can inhibit multiple levels of the ascending pain pathways for pain signal transmission in supraspinal regions related to nociceptive message integration and participate in facilitatory and inhibitory descending pathways [13,38,39]. For example, in the spinal cord, the release of presynaptic opioids activates postsynaptic opioid receptors and promotes the opening of postsynaptic K^+^ and/or Cl^−^ channels, evoking hyperpolarization inhibition potentials in dorsal horn neurons [13,38]. Similarly, the top-down regulation of the opioid receptor system within the periaqueductal gray (PAG) and the rostral ventral medulla (RVM) can exert descending regulatory control on the transmission of nociceptive signals [40,41], showing the importance of the function and quantity of opioid receptors in the process of nociceptive information transmission and inhibition. Despite their many serious side effects, including constipation, nausea, vomiting, respiratory depression, sedation, dysphoria, addiction, and tolerance [14,42,43], opioids have long been the most widely used analgesics and are considered the standard of care for the treatment of various pain states due to their high efficacy and availability [43,44]. From a clinical point of view, opioid-induced hyperalgesia and analgesia tolerance are noteworthy, as the management of such patients during care is a challenge, which also means that whether or not to use opioids in the clinical setting should be approached with caution [38,39,42,44,45].

### 2.2. Opioid Receptor-Targeted Drugs for NP

The development of opioid receptor-targeted drugs has long been the focus of analgesic drug research. As the main receptor of the most important analgesic target and related side effects of opioids [14,46,47], MOR remains the center of these studies [48]. Almost all opioid drugs used in the clinical setting are agonists of MOR, and the representative classical opioid analgesics such as morphine and fentanyl show high selectivity to MOR [43,44,46,47,48,49]. Most analgesics have limited efficacy for NP; still, opioids can provide short-term but substantial pain relief for moderate to severe pain of neuropathic origin [43,50], as proven by a large number of clinical trials [51]. Nevertheless, in most treatment guidelines for NP, opioids (e.g., morphine and oxycodone) are only suggested for use as third-line treatment drugs because of drug safety considerations and their low analgesic effect in NP [49,52,53,54]. In this context, multitargeted treatment of opioid receptors can improve the drug tolerance and safety profile [55,56], and previous studies have shown that simultaneous activation of MOR and DOR with bivalent agonist ligands delivers potent analgesia in experimental NP [57,58,59]. For example, the benzomorphan ligand LP2, a multitarget MOR-DOR agonist, can significantly ameliorate mechanical allodynia from the early phase of treatment up to 21 days post ligature in unilateral sciatic nerve chronic constriction injury in male Sprague–Dawley (SD) rats [57]. Moreover, RV-Jim-C3, a newly synthesized bifunctional opioid agonist derived from combined structures of fentanyl and enkephalin in rodents with high-affinity binding to both MOR and DOR, demonstrates the potent-efficacious activity of analgesia and antihyperalgesic [58]. Another multitarget MOR-DOR agonist, SRI-22141, was tested in the clinically relevant models of HIV neuropathy and chemotherapy-induced peripheral neuropathy, in which it was found that while providing equal or enhanced efficacy versus morphine, the tolerance was significantly reduced, with almost non-dependence during the course of repeated administration [59]. In addition, it was found that a multifunctional peptide, DN-9, can activate peripheral KOR and MOR via subcutaneous injection and dose-dependently produce antinociception, exhibiting a more potent ability than morphine, reducing the typical side effects of MOR activation [60]. Hence, some drug development focuses on peripherally restricted KOR agonists to retain analgesic efficacy with a more favorable adverse effect [61]. On the other hand, a mixed MOR agonist and DOR-KOR antagonist were reported to have potent antinociceptive activity and diminished rodent tolerance and dependence [62]. The results of these previous studies suggest that multimodal targeting at the opioid receptors has tremendous potential to enhance the required therapeutic effects for NP.

Furthermore, the analgesic effect of opioids was once thought to be mediated by activation of the downstream inhibitory Gαi/o protein signaling pathway, and its side effects were thought to be caused by the β-arrestin signal pathway [12,13,15]. Extensive research has been invested in the development of G protein/β-arrestin-biased MOR agonists to find efficient and low-toxicity targeted MOR analgesic drugs [63,64] while simultaneously solving the pathological changes in opioid receptors related to NP, then enhancing the therapeutic effect on NP.

### 2.3. The Same Outcome of NP and Long-Term Opioid Exposure

Compared with other types of pain, the efficiency of opioids in NP treatment is still low, which might be because the occurrence of the latter shares the same neuropathological consequences with continuous and/or repeated opioid exposure, which is manifested by hyperalgesia and opioid tolerance [10,11,42,43,48,65]. In other words, it is plausible that the mechanisms that promote the above two phenomena might synergize and ultimately aggravate NP when patients are treated with opioids [48,65,66], which may be useful to address the failure of strong opioids in NP, the commonness of which is reflected in similar changes at the tissue level, remarkably showing the downregulation of opioid receptors in the central nervous system (CNS) [67] and peripheral nervous system (PNS) [68] accompanied by increased expression of pronociceptive neurotransmitters (e.g., cholecystokinin, calcitonin gene-related peptide, substance P, etc.) [69] and chemokines [70]. In addition, interestingly, the exhibited circulating levels of β-endorphin in patients with chronic lower back pain are low [71,72]. On the contrary, the expression of dynorphin [73,74] and nociception [75] in multiple supraspinal sites was significantly increased under NP conditions.

A large number of experiments have proven that opioids have significantly lower efficacy in allodynia and hyperalgesia caused by NP and are more prone to produce analgesic tolerance [76,77,78,79,80,81]. Thus, the administration of strong opioids may worsen rather than alleviate NP [82,83]; for instance, morphine treatment may aggravate pre-existing allodynia in animals with peripheral injury [83] and prolong the duration of allodynia after treatment ends [84,85]. Based on this, some mechanistic explanations have been proposed for the loss of strong opioid effectiveness in NP conditions, including the release of dynorphin [67], the downregulation of MOR in the spinal cord [76,77] and dorsal root ganglion (DRG) [78,79], the increase in the serotonin level [80], increased methylation of the MOR gene promoter in DRG and/or the stimulation of glutamate receptor [81], the decrease in morphine concentration in the brain [86], etc. All of these factors collectively lead to NP patients needing higher doses of morphine than non-NP patients to achieve effective analgesia.

## 3. Downregulation of Opioid Receptors after Repeated Opioid Treatment

Long-term repeated use of opioids, such as morphine, can lead to analgesic tolerance in patients, which is a major factor that reduces the opioid effect and is also one of the important reasons limiting application in various pain-related diseases [14,42,43,50]. The mechanism underlying opioid tolerance mainly focuses on intracellularly adaptive changes in cells containing opioid receptors, including the desensitization of receptor signals, downregulation of opioid receptors, compensatory/antagonistic processes, etc. [15,68,87]. Although research on the mechanism of opioid tolerance is still in full swing, there is no doubt that the imbalance of the desensitization–internalization–resensitization of MOR plays a crucial role [15], which is also the main focus of our review. As an important component directly involved in opioid analgesia, MOR undergoes conformational changes after being activated by agonists, leading to G-protein activation and mediating a series of intracellular activities (Figure 1a). Subsequently, the intracellular domain of MOR is phosphorylated by G-protein-coupled receptor kinases and/or second messenger-regulated protein kinases. After phosphorylation, β-arrestin 2 is recruited to the plasma membrane to bind with MOR and prevent the further activation of G protein to realize desensitization [87], accelerating the uncoupling of MOR with G protein, forming receptor–protein complexes (MOR/β-arrestin 2/Adaptin-2/Clathrin) at the same time to promote the internalization of MOR until the internalization is suspended by dynamin [21,87,88,89]. There are two consequences of MOR after internalizing into the cell: one is degraded by the lysosome, and another is returned to the cytomembrane to continue its function, which causes desensitization [21,88].

The internalization process is important for maintaining the normal quantity and physiological function of membranous MOR; when it is interfered with and is out of order, e.g., due to repeated use of opioids, this leads to a reduction in functional MOR on the cell membrane, causing opioid tolerance and/or hyperalgesia [21,88,89]. After long-term use of morphine, although the intensity of internalization did not decrease, the recovery of MOR after endocytosis decreased, leading to morphine tolerance [88,90]. It has been found that opioids without enhanced internalization, such as morphine, are more likely to cause opioid tolerance, which also suggests that receptor internalization can reduce the occurrence of tolerance; in contrast, methadone and the selective peptide agonist [D-Ala^2^, N-Me-Phe^4^, Gly^5^-ol]-enkephalin (DAMGO) produce marked receptor endocytosis with little tolerance [21,88,91,92,93]. In addition, some research found that the development of morphine tolerance in gene mice with β-arrestin 2 deletion is delayed [90,94,95], which seems to contradict the aforementioned relationship between internalization and tolerance. However, some studies have proven that β-arrestin 2 mediates acute rather than chronic opioid tolerance, which indicates that opioid-induced antinociceptive tolerance may even occur without the activation of β-arrestin 2 [96], as it should be; further research is needed to confirm these conjectures.

Based on the above facts, the following general explanation for tolerance caused by internalized imbalance is proposed. The low tolerance of opioids with strong internalization is due to the periodicity of internalization, promoting the dephosphorylation of MOR in endosomes; then, the active receptors can be recycled to the surface of the cytomembrane [90,95]. It is difficult for morphine to impel the internalization of desensitized MOR, and the latter cannot be dephosphorylated within the cell [88], so repeated morphine treatment leads to the accumulation of desensitized MORs, representing the downregulation of functional MOR and ultimately leading to tolerance.

## 4. Downregulation of Opioid Receptors in DRG following Nerve Injury

Nerve injury is a common cause of NP [10,11]. The change in neuronal signal transduction caused by peripheral nerve injury is an important pathological basis for the occurrence and continuous progress of NP, in which the abnormal expression of pain sensors is the main factor, especially the change in the expression of sensory neurons in the DRG of the pain upload pathway [97,98]. The abnormal expression of opioid receptors under NP status will be discussed here. It has been reported that a time-dependent decrease in MOR at either mRNA and/or protein levels was observed in trauma-related DRGs following peripheral nerve injury [78,99,100,101]. For example, early studies found that peripheral axotomy can reduce the number and intensity of MOR-positive neurons and MOR-like immunoreactivity in DRG of SD rats and monkeys [100]. In addition, after L5 spinal nerve ligation (SNL), MOR mRNA and protein in the ipsilateral L5 DRG of male SD rats decreased in a time-dependent manner on days 3–14, and the change in MOR protein even lasted until the 35th day [78]. Similarly, the downregulation of MOR mRNA in ipsilateral DRG was observed on days 3 and 14 after sciatic nerve ligation in male Wistar rats [101]. Moreover, compared with naive animals, the MOR level in the DRG of rats following chronic constriction injury (CCI) of the sciatic nerve increased by 110%, but the KOR and DOR remained the same [102]. Changes in opioid receptor expression, especially MOR, reduce the inhibitory function and efficiency of endogenous opioid peptides when the body responds to nociceptive stimuli, enhances neuronal excitability, and promotes the occurrence of sensory sensitization and the development of NP [81,103]. Although the mechanisms of the downregulation of opioid receptors in DRG neurons after peripheral nerve injury are still unintelligible, additional shreds of evidence revealed that this downregulation occurs at multiple levels, with changes in related key molecules affecting the number and function of opioid receptors (Table 2) and involving different molecular mechanisms of NP.

### 4.1. Epigenetic Level

DNA methylation is a type of epigenetic modification process that is mainly initiated by DNA methyltransferases (DNMTs), including DNMT1, DNMT3a, and DNMT3b, which can inhibit gene transcriptional expression by acting as a docking site for transcription repressors and/or physically blocking the binding of transcription factors [120].

DNMT1 and DNMT3a are expressed in DRG neurons and participate in the production and maintenance of NP due to the increase in mRNA and/or protein in trauma-related DRG neurons after peripheral nerve injury [104,121]. DNMT3a was reported to be significantly increased in injury-related DRGs after unilateral L5 SNL and in the CCI of the unilateral sciatic nerve in male SD rats [104,121]. Inducing an increase in the methylation of the *Kcna2* gene (important potassium [K^+^] channel gene, encoding Kv1.2) and the promoter of the 5′-untranslated region (UTR) of *Oprm1* and *Oprk1* genes reduces MOR and KOR expression in injury-related DRG neurons and further enhances peripheral sensitization [104,105].

Methyl-CpG-binding domain protein 1 (MBD1), a methylated DNA-binding protein and an epigenetic repressor for the regulation of gene transcriptional activity, is critical for the genesis of NP because DRG MBD1-deficient mice exhibit reduced responses to blunted nerve-injury-induced pain hypersensitivities; additional results suggest that MBD1 may participate in the occurrence of NP by regulating DNMT3a to regulated *Kcna2* and *Oprm1* gene expression in the DRG neurons [106]. Ten-eleven translocation methylcytosine dioxygenase 1 (TET1), which can induce DNA demethylation, showed that overexpression in DRGs of rats can significantly alleviate L5 SNL-induced pain hypersensitivities by rescuing the expression of MOR and Kv1.2 during the development and maintenance periods while restoring morphine analgesia and attenuating morphine analgesic tolerance development [107].

Histone methylation, a type of histone modification, can regulate gene expression. The euchromatic histone-lysine N-methyltransferase 2 (G9a) encoded by the *Ehmt2* gene can downregulate the expression of *Kcna2*, *Oprm1*, *Oprk1*, and *Oprd1* genes in injury-related DRG neurons in a preclinical mouse model of unilateral L4 SNL [108,109] or unilateral L5/6 SNL of SD rats [110]. In addition, the histone–lysine methyltransferase SUV39H1, which is mainly expressed in small DRG neurons, can participate in the occurrence of NP through epigenetic silencing of MOR expression in DRGs following unilateral L5 SNL of SD rats [111].

### 4.2. Transcription Level

As important regulators in the process of gene expression, long non-coding RNAs (lncRNAs) have been extensively studied to regulate the occurrence and development of chronic pain [122]. DRG-specifically enriched lncRNA (DS-lncRNA) highly expressed in DRG neurons was significantly downregulated following L4 SNL, CCI, or axotomy of the unilateral sciatic nerve, increasing the expression of *Ehmt2*/G9a triggered by the transcription cofactor RALY/RNP II, which correspondingly reduced the expression of opioid receptors and *Kcna2* in related DRGs, contributing to NP [112].

Transcription factors (TFs) are a large class of sequence-specific DNA-binding proteins and act as key regulators of gene transcription. Neuron-restrictive silencer factor (NRSF) was significantly upregulated in the injury-related DRGs of partial ligation of the sciatic nerve [113] and sarcoma-inoculated murine model [114], combining with the targeted *Oprm1* gene in a sequence-specific manner and reducing the expression of the latter via HDAC-involved mechanisms [113], which is one of their main functions in regulating gene transcription. Furthermore, CCAAT/enhancer binding protein β (C/EBPβ) was found to be upregulated in the ipsilateral L3/4 DRG neurons of the CCI model and significantly reduced the amounts of *Oprm1* mRNA and MOR protein after the consensus binding motif within the promoter region of the *Ehmt2* gene [115]. Octamer transcription factor 1 (OCT1) is also a transcription activator, and its binding motif was found in the promoter region of the *Dnmt3a* gene [104]; the level of OCT1 protein was time-dependently increased in the ipsilateral L4/5 DRGs of rats after CCI of the sciatic nerve, which is responsible for the upregulation of *Dnmt3a* mRNA and protein and the downregulation of *Oprm1* mRNA and MOR protein triggered by *DNMT3a*, to promote the development of NP [116].

### 4.3. Post-Transcriptional Level

MicroRNAs (miRs) are a class of endogenous small non-coding RNAs that can directly and/or indirectly participate in the progression of pain [122]; miR-143 can directly target and inhibit *Dnmt3a* promoter activity [123]. It was reported that peripheral nerve injury caused by L5 SNL reduced the expression of miR-143 in related DRGs, leading to the upregulation of *Dnmt3a* expression and damaging the expression of *Oprm1* mRNA and MOR [117].

RNA modification is a typical process regulating gene expression at the post-transcriptional level; the most common modification is N^6^-methyladenosine (m^6^A) modification of RNA [124]. The high expression of m^6^A demethylase fat mass in the ipsilateral L5 DRG after SNL or in the ipsilateral L4/5 DRGs after CCI in rats erases m^6^A in *Ehmt2* mRNA, increases the expression level of G9a in DRGs, and causes NP [118]. Eukaryotic initiation factor 4F (eIF4F) is one of the protein complexes involved in the initiation of protein translation and is also a key effector regulating gene expression at the post-transcriptional level [125]. It was reported that eIF4G2 (one of three isoforms in eIF4F) is co-expressed with MOR in small DRG neurons, the expression of which in the ipsilateral L4 DRG of mice increased over time after SNL, reducing the expression of MOR [119], which suggests that eIF4G2 may promote NP by negatively regulating the expression of MOR in injury-related DRG neurons [119].

Contrary to previous studies, interestingly, it was observed that NOR upregulation in the DRGs in NP animal models with partial sciatic nerve transection and sciatic nerve ligation [75,126] may increase the analgesic properties of endogenous ligand nociceptin under NP conditions without the opioid-related side effects [127]. However, whether or not there is a close relationship between these mechanisms and how they may be combined to cause the downregulation of opioid receptors, especially MOR, under nerve-injury conditions still need further research. In addition, the pain information transmission caused by the original peripheral nerve injury is out of balance when the expression of opioid receptors is downregulated in DRGs, resulting in the inability of endogenous opioid peptides to act on the receptor and leading to peripheral sensitization, which is also an important mechanism for the formation of morphine tolerance that seems to indicate that subcutaneous morphine injection in NP patients is difficult to modify to achieve better efficacy [77,81,105,108,128].

## 5. Downregulation of Opioid Receptors in the CNS following Nerve Injury

There is still some controversy about NP leading to the up- or downregulation of opioid receptor expression in the spinal cord. It was observed in the saphenous partial ligation model that the expression of MOR in the ipsilateral paw skin, L3/4 DRGs, and spinal cord was significantly increased [129]. However, functional downregulation and/or desensitization of MOR has also been observed in the spinal dorsal horn, particularly in laminae I and II, in nerve-injury neuropathy [101,130] and diabetic neuropathy [131,132], although with no significant decrease in expression [78,130,131,132]. In contrast, further studies showed that the density of MOR-like immunoreactivity is significantly reduced in the ipsilateral dorsal horn (laminae I and II) L4/5 segment after L4/5 SNL in SD rats [99], and similar studies have found that the expression of MOR protein in the L4/5 spinal cord segment decreased from the 35th day of ligation in the L5 SNL model of rats [100]. There is a lack of available functional spinal MOR due to this downregulation/desensitization, which leads to a lower analgesic effect of intrathecal injection of opioids compared with intraperitoneal injection and failure to reduce allodynia following neuropathy [130,131,132].

As early as more than ten years ago, some clinical studies reported a significant decrease in opioid receptor binding in supraspinal regions (e.g., posterior midbrain; anterior cingulate; medial thalamus; and the insular, temporal, and prefrontal, inferior parietal cortices, etc.) of NP patients by [^11^C]-diprenorphine binding and positron emission tomography (PET) technology (Figure 2) [133,134]. Since then, similar results related to the reduced availability of opioid receptors in the supraspinal regions in CCI to the sciatic nerve [135] and the spared nerve injury model [136] have been reported; researchers have begun to agree that the reduced expression of opioid receptors may be the direct result of chronic pain rather than individual differences [135,136,137]. It is also recognized that opioid receptors expressed in the CNS are much more likely to be downregulated after nerve injury than those expressed in peripheral afferent fibers (including C, Aβ, and Aδ afferent fibers) [77,137,138]. This situation is also accompanied by the altered efficacy of G-protein stimulation in opioid-sensitive CNS structures and a reduction in the expression of several opioid receptor mRNAs, which indicates that NP can recruit and modify several opioidergic circuits in the brain [139]. In short, the functional downregulation or desensitization of the opioid receptor in the brain under the NP state may contribute to the reduction in opioid drug efficacy in the brain of animal models of NP [140,141].

## 6. Conclusions

As a common pain syndrome, NP causes physical pain and psychological torture, producing tremendous social and economic burdens. Therefore, although the pathogenesis of NP is so complex that it has not been revealed clearly, it is imperative to explore better methods for controlling symptoms and even curing them completely. Opioids are the last choice for the drug treatment of NP when other drugs gradually lose their efficacy; however, about two-thirds of NP patients report poor pain control [142], which necessitates the repeated and prolonged use of opioids and higher doses to alleviate pain, such that various side effects caused by opioids cannot be avoided. The decrease in the therapeutic effect of opioids can be explained by the commonality of the downregulation of opioid receptors in both the PNS and/or CNS under NP status and the long-term, repeated use of opioids (Figure 1). This review reveals this commonality, and it is obvious that the aspects of this commonality do not inhibit each other but interact with each other, forming a vicious circle of the continuous downregulation of opioid receptors caused by opioid treatment under NP conditions. Therefore, future research should target these underlying mechanisms to develop drugs that exert antinociceptive effects, maintain opioid analgesia, and resist opioid analgesic tolerance in the management of NP. Unfortunately, few studies have been conducted on the effects of the heavy use of opioids and/or NP on the opioid receptor function in the CNS, which is worthy of attention and investigation. It should also be noted that some key factors of the mechanism summarized in this review (such as DNMTs, G9a, TFs, etc.), especially internalization-related β-arrestins, are also expressed in other cells besides neurons and participate in various normal physiological activities. These possible side effects should be considered when therapeutics for NP are developed in the future.

## Figures and Tables

**Figure 1 ijms-24-05981-f001:**
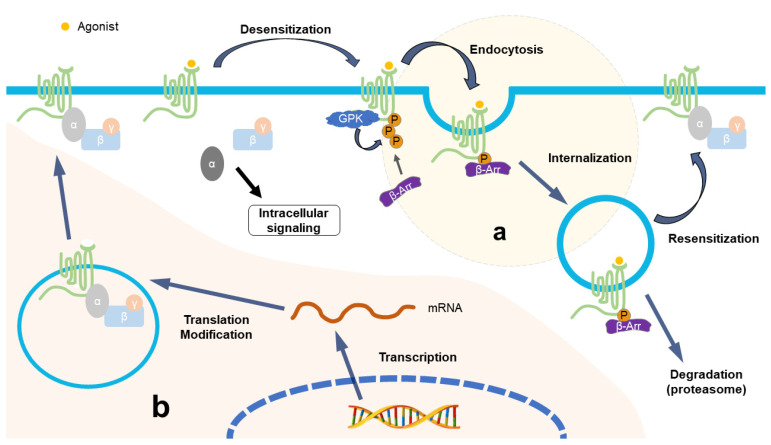
Identical consequences of NP and repeated opioid treatment from the perspective of opioid receptors. (**a**) Repeated use of opioid drugs promotes the imbalance of the desensitization–internalization–resensitization cycle of opioid receptors; for example, morphine cannot cause the internalization of MOR, leading to the accumulation of deactivated MOR on the cell membrane, resulting in the downregulation of opioid receptor function. The treatment of NP patients with opioids aggravates the downregulation of opioid receptors in the form of a vicious circle and aggravates the symptoms of opioid tolerance and hyperalgesia. (**b**) In the NP state, various factors affect the synthesis of opioid receptors at the epigenetic level, transcriptional level, and post-transcriptional level, leading to a reduction in opioid receptors.

**Figure 2 ijms-24-05981-f002:**
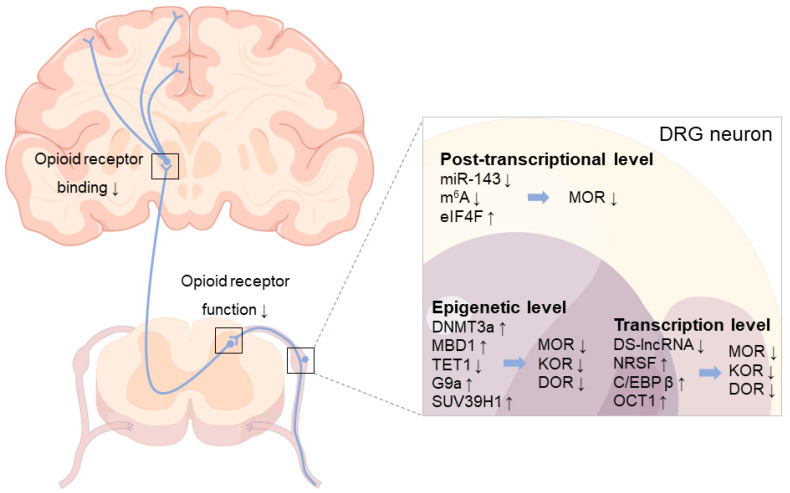
A multidimensional view of opioid receptor downregulation in the NP state. After peripheral nerve injury, opioid receptors in DRG neurons are regulated at epigenetic, transcriptional, and post-transcriptional levels through various mechanisms, leading to the downregulation of opioid receptors and promoting the development of NP. Moreover, the decrease in opioid receptor binding in some brain regions (e.g., posterior midbrain; anterior cingulate; medial thalamus; and the insular, temporal, prefrontal cortices, inferior parietal cortices, etc.) and the downregulation of opioid receptors in the spinal dorsal horn cord can be observed following NP. Arrows indicate changes in gene or receptor expression, as well as decreased opioid receptor function and binding.

**Table 1 ijms-24-05981-t001:** Effects of classical opioid receptors.

**MOR**
**Central**
Analgesia Constipation Euphoria Respiratory depression
**Peripheral**
Analgesia Constipation Reduced inflammation
**DOR**
**Central**
Analgesia Anxiolysis Convulsions
**Peripheral**
Analgesia Constipation
**KOR**
**Central**
Analgesia Diuresis Dysphoria
**Peripheral**
Analgesia Reduced inflammation

**Table 2 ijms-24-05981-t002:** Mechanisms of opioid receptor downregulation in DRG following peripheral nerve injury.

	Changes in Key Factors		Relative Targets	Changes in Opioid Receptors	Efficacy of Opioids after Blocking Changes	References
**Epigenetic level**						
DNMTs	DNMT3a	↑	*Oprm1* and *Oprk1*	MOR ↓ KOR ↓	↑	[104,105]
MBDs	MBD1	↑	*Oprm1*	MOR ↓	↑	[106]
TETs	TET1	↓	*Oprm1*	MOR ↓	↑	[107]
Histone methyltransferase	G9a	↑	*Oprm1*, *Oprk1*, and *Oprd1*	MOR ↓ KOR ↓ DOR ↓	↑	[108,109,110]
	SUV39H1	↑	*Oprm1*	MOR ↓	Untested	[111]
**Transcription level**						
LncRNA	DS-lncRNA	↓	*Ehmt2*/G9a	MOR ↓ KOR ↓ DOR ↓	Untested	[112]
TFs	NRSF	↑	*Oprm1*	MOR ↓	↑	[113,114]
	C/EBP β	↑	*Ehmt2*/G9a	MOR ↓	↑	[115]
	OCT1	↑	*Dnmt3a* mRNA	MOR ↓	↑	[116]
**Post-transcriptional level**						
MicroRNA	miR-143	↓	*Dnmt3a* mRNA	MOR ↓	↑	[117]
RNA modification	m^6^A	↓	*Ehmt2* mRNA	MOR ↓	↑	[118]
Protein complex	eIF4F	↑	MOR	MOR ↓	Untested	[119]

DNMTs, DNA methyltransferases; *Oprm1*, μ-opioid receptor gene; *Oprk1*, κ-opioid receptor gene; *Oprd1*, δ-opioid receptor gene; MBDs, methyl-CpG-binding domain proteins; TETs, ten-eleven translocation methylcytosine dioxygenases; G9a, euchromatic histone-lysine N-methyltransferase 2; SUV39H1, variegation 3-9 homolog 1; LncRNA, long non-coding RNA; DS-lncRNA, DRG-specifically enriched lncRNA; *Ehmt2*, gene encoding histone methyltransferase G9a; TFs, transcription factors; NRSF, neuron-restrictive silencer factor; C/EBP β, CCAAT/enhancer binding protein β; OCT1, octamer transcription factor 1; m^6^A, N^6^-methyladenosine; eIF4F, eukaryotic initiation factor 4F. Arrows indicate changes in the expression of key factors such as genes or receptors, as well as changes in the efficacy of opioids after blocking the aforementioned changes.

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
