# Peer review of "The Downregulation of Opioid Receptors and Neuropathic Pain"

_ijms, 2023, doi:10.3390/ijms24065981_

Round 1

Reviewer 1 Report (Previous Reviewer 2)

The manuscript has now improved. Unfortunately, there are still many language problems (syntax, grammar, vocabulary, missing words). Extensive language editing is still needed (e.g. by a professional service). Some statements are trivial, e.g. "... patients with NP are significantly more likely to receive opioids than non-NP" on line 100.

Author Response

Dear Editor and Reviewers,

Thanks very much for taking the time to review this manuscript. I appreciate all your valuable comments and suggestions! We have carefully considered these suggestions and make some supplements and changes to this manuscript as required. We have tried our best to improve and make some modifications, and have conducted extensive English editing through the English editing service provided by MDPI.

Please find my itemized responses below (point-to-point) and my revisions/corrections in the re-submitted files.

Thanks again!

Best regards,

Lin Li

-------------------------

Review Report (Reviewer 1)

The manuscript has now improved. Unfortunately, there are still many language problems (syntax, grammar, vocabulary, missing words). Extensive language editing is still needed (e.g. by a professional service). Some statements are trivial, e.g. "... patients with NP are significantly more likely to receive opioids than non-NP" on line 100.

We apologize again for the poor language of our manuscript. We have conducted extensive English editing of this manuscript to improve its readability through the English editing service provided by MDPI, and the trivial statement had been modified at the same time (Page 4, lines 121-123).

Reviewer 2 Report (Previous Reviewer 1)

Authors appropriately responded or answered to my comments (I guess the info. on the pages and lines (page3, lines 78-111) that addresses my comment #3 is incorrect. I assume that the authors added section 2.2 (page3, lines 92-133) to answer my comments). I think the revised manuscript is now suitable for publication in the International Journal of Molecular Sciences.

Author Response

Dear Editor and Reviewers,

Thanks very much for taking the time to review this manuscript. I appreciate all your valuable comments and suggestions! We have carefully considered these suggestions and make some supplements and changes to this manuscript as required. We have tried our best to improve and make some modifications, and have conducted extensive English editing through the English editing service provided by MDPI.

Please find my itemized responses below (point-to-point) and my revisions/corrections in the re-submitted files.

Thanks again!

Best regards,

Lin Li

-------------------------

Review Report (Reviewer 2)

Authors appropriately responded or answered to my comments (I guess the info. on the pages and lines (page3, lines 78-111) that addresses my comment #3 is incorrect. I assume that the authors added section 2.2 (page3, lines 92-133) to answer my comments). I think the revised manuscript is now suitable for publication in the International Journal of Molecular Sciences.

I think you are right. I'm sorry about that maybe we didn't make it clear in our last reply. Finally, thank you for your high recognition

Reviewer 3 Report (New Reviewer)

In this manuscript, Li et al., summarize literature about opioid receptor expression during neuropathic pain in both peripheral and central nervous systems. Opioids and neuropathic pain are interesting topics for scientists, physicians and the general public, thus this manuscript should have a wide readership. However, the manuscript needs to be significantly improved. Below are my major concerns:

  1. There are many long sentences that are hard to understand. The author should spend more effort to improve English writing. Short and clear sentences will be easier for readers to follow.

  1. The manuscript focuses on the regulation of opioid receptors, but not opioid peptides, in neuropathic pain. The endogenous opioid system contains both opioid peptides and receptors, if opioid receptors are highly regulated in neuropathic pain, how about the opioid peptides? The manuscript would be more intriguing if the authors could include how the opioid peptides are regulated in neuropathic pain, which could give readers the whole picture of how the endogenous opioid system is regulated during neuropathic pain. Otherwise, it would be better if the authors could introduce the endogenous opioid system and its role in pain perception in the introduction.

  1.  Opioids-induced analgesia is mainly mediated by the µ-opioid receptor (MOR), which is consistent with the table and reference in the manuscript. Dor and Kor may have an analgesic effect, but this is not clear. Thus, it would be more specific and straightforward if the authors could focus on MOR expressions/modifications in neuropathic pain, and include the opioid peptides (suggested in major concern 2).

I also have many minor concerns, but it may not be worth listing here now before the authors revised their manuscript.

Author Response

Dear Editor and Reviewers,

Thanks very much for taking the time to review this manuscript. I appreciate all your valuable comments and suggestions! We have carefully considered these suggestions and make some supplements and changes to this manuscript as required. We have tried our best to improve and make some modifications, and have conducted extensive English editing through the English editing service provided by MDPI.

Please find my itemized responses below (point-to-point) and my revisions/corrections in the re-submitted files.

Thanks again!

Best regards,

Lin Li

-------------------------

Review Report (Reviewer 3)

In this manuscript, Li et al., summarize literature about opioid receptor expression during neuropathic pain in both peripheral and central nervous systems. Opioids and neuropathic pain are interesting topics for scientists, physicians and the general public, thus this manuscript should have a wide readership. However, the manuscript needs to be significantly improved. Below are my major concerns:

  1. There are many long sentences that are hard to understand. The author should spend more effort to improve English writing. Short and clear sentences will be easier for readers to follow.

We apologize again for the poor language of our manuscript. We have conducted extensive English editing of this manuscript to improve its readability through the English editing service provided by MDPI.

  1. The manuscript focuses on the regulation of opioid receptors, but not opioid peptides, in neuropathic pain. The endogenous opioid system contains both opioid peptides and receptors, if opioid receptors are highly regulated in neuropathic pain, how about the opioid peptides? The manuscript would be more intriguing if the authors could include how the opioid peptides are regulated in neuropathic pain, which could give readers the whole picture of how the endogenous opioid system is regulated during neuropathic pain. Otherwise, it would be better if the authors could introduce the endogenous opioid system and its role in pain perception in the introduction.

We agree with you, but we prefer to focus on opioid receptors rather than opioid peptides in the limited article space, so we have added a little knowledge in this field, including changes in the endogenous opioid system (Section 2.1, pages 2-3, lines 71-113) and opioid peptides (Page 4, lines 168-172).

  1. Opioids-induced analgesia is mainly mediated by the µ-opioid receptor (MOR), which is consistent with the table and reference in the manuscript. Dor and Kor may have an analgesic effect, but this is not clear. Thus, it would be more specific and straightforward if the authors could focus on MOR expressions/modifications in neuropathic pain, and include the opioid peptides (suggested in major concern 2).

I also have many minor concerns, but it may not be worth listing here now before the authors revised their manuscript.

Thanks for your advice, we have added some information about the opioid peptides (Page 4, lines 168-172) and increased the degree of attention to the expression/modifications of MOR appropriately at the same time (Section 4, pages 6-9).

Round 2

Reviewer 1 Report (Previous Reviewer 2)

The language has now imporoved. There are still some redundant sentences.

Author Response

Dear Editor and Reviewers,

Thank you for your valuable comments on my manuscript many times during your busy schedule! We have carefully read and adopted these suggestions, and we have made our best efforts to further revise this manuscript. At the same time, we have conducted extensive English editing again through the English editing service provided by MDPI.

Thanks again!

Best regards,

Lin Li

-------------------------

Review Report (Reviewer 1)

Comments and Suggestions for Authors

The language has now improved. There are still some redundant sentences.

I agree with you, so I deleted some redundant descriptions in the hope of making my manuscript more concise to read. The attached manuscript is in review mode of Word using the “Track Changes” function.

Reviewer 3 Report (New Reviewer)

Thanks for the updated manuscript. The manuscript has been improved, but two concerns remain:

  1. English writing. The authors claimed they used an English editing service to improve the readability of this manuscript. However, I did not see obvious changes.  For example, the authors modified the location of a few words in the abstract, but most sentences are still too long and hard to read.

  1. Figure 2. The illustration is misleading. If the blue line represents modified neural circuits during nerve injury, does this mean that nerve injury changes gene expression in ipsilateral DRG and brain, but contralateral spinal cord? This is not how the somatosensory pathway works. 

I would be very helpful if the authors could find a native speaker to proofread the manuscript. Besides that, I have no other concerns.

Author Response

Dear Editor and Reviewers,

Thank you for your valuable comments on my manuscript many times during your busy schedule! We have carefully read and adopted these suggestions, and we have made our best efforts to further revise this manuscript. At the same time, we have conducted extensive English editing again through the English editing service provided by MDPI.

Thanks again!

Best regards,

Lin Li

-------------------------

Review Report (Reviewer 3)

Comments and Suggestions for Authors

Thanks for the updated manuscript. The manuscript has been improved, but two concerns remain:

  1. English writing. The authors claimed they used an English editing service to improve the readability of this manuscript. However, I did not see obvious changes.  For example, the authors modified the location of a few words in the abstract, but most sentences are still too long and hard to read.

I agree with you, so I have made some modifications to the manuscript, including some long sentences, to make it easier to read and understand. The attached manuscript is in review mode of Word using the “Track Changes” function.

  1. Figure 2. The illustration is misleading. If the blue line represents modified neural circuits during nerve injury, does this mean that nerve injury changes gene expression in ipsilateral DRG and brain, but contralateral spinal cord? This is not how the somatosensory pathway works. 

I would be very helpful if the authors could find a native speaker to proofread the manuscript. Besides that, I have no other concerns.

Thank you for your reminder. We have made some adjustments to Figure 2 to avoid misleading the reader.

English-Editing-Certificate-62365

This manuscript is a resubmission of an earlier submission. The following is a list of the peer review reports and author responses from that submission.

Round 1

Reviewer 1 Report

The manuscript by Li et al. nicely reviews/introduces the down-regulation of the opioid receptors in DRG of the neuropathic pain (NP)models and suggests possible mechanisms that will affect the membrane expression of opioid receptors in the NP condition. I feel this is a novel topic nowadays, important for the pain management, and consequently the present review will be welcomed by the journal's audience.

I just have few comments as below.

Specific comments

1. It is not clear whether the down-regulation of the opioid receptors surely affect the pharmacological effect of opioids in the animal models that the authors introduce in this review. Authors had better include some information on the levels of decrease of opioid receptors and the levels of decrease in the therapeutic effects (i.e., antinociceptive effect) of opioid (e.g., in the table). If there are some positive/negative correlations, it would be interesting.

2. In figure 1, authors had better include the info. on the "opioid receptors" per se (i.e., what type of receptors will decrease in the DRG?).

3. I think it is known that the destiny of the receptor (i.e., they will be down regulated by the degradation or they will be recovered by the recycling) will be changed by the ligands or therapeutic medicines (it may be very important discuss on the beta-arrestin bias of the ligands). If authors could add some information on this in the neuropathic pain models, it would extend the impact of this review for the readers.

Author Response

Dear Editor and Reviewers,

Thanks very much for taking the time to review this manuscript. I appreciate all your valuable comments and suggestions! We have carefully considered these suggestions and make some supplements and changes to this manuscript as required. We have tried our best to improve and made some revisions.

Please find my itemized responses below (point-to-point) and my revisions/corrections in the re-submitted files.

Thanks again!

Best regards,

Lin Li

-------------------------

Reviewer 1 report:

  1. It is not clear whether the down-regulation of the opioid receptors surely affect the pharmacological effect of opioids in the animal models that the authors introduce in this review. Authors had better include some information on the levels of decrease of opioid receptors and the levels of decrease in the therapeutic effects (i.e., antinociceptive effect) of opioid (e.g., in the table). If there are some positive/negative correlations, it would be interesting.

Thank you for your suggestion. We have added the corresponding results in Table 1 (Page 6, line 217).

  1. In figure 1, authors had better include the info. on the "opioid receptors" per se (i.e., what type of receptors will decrease in the DRG?).

Thanks for your suggestion for revising the figure, and the precedent version of this figure has been replaced (Figure 2, Page 9, line 347).

  1. I think it is known that the destiny of the receptor (i.e., they will be down regulated by the degradation or they will be recovered by the recycling) will be changed by the ligands or therapeutic medicines (it may be very important discuss on the beta-arrestin bias of the ligands). If authors could add some information on this in the neuropathic pain models, it would extend the impact of this review for the readers.

We agree with you suggestion, so that we added the content about opioids in neuropathic pain treatment in the manuscript (Page 3, lines 78-111).

It seems that only one file can be uploaded here, so I can only upload the Marked-up version instead of the Clean copy. Please see the attachment.

Reviewer 2 Report

The text is very difficult to follow. Extensive language corrections are necessary throughout the manuscript. The English needs to be checked by a native speaker.

The review is not focused on one specific topic. It rather is a combination of two different reviews: one on neuropathic pain and one on opioid receptors. Both topics are discussed poorly. Large parts consist of unnecessary detailed information on well-known topics (e.g. receptor internalization), while others lack important information such as the mechanisms underlying opioid tolerance (e.g. line 218).

A connection between opioid receptors and neuropathic pain is described. However, endogenous opioids or opioid drugs are not discussed. Central or peripheral side effects of opioids  (the main problem limiting opioid treatments in the clinical setting) are also not discussed in detail.

A few examples for specific changes are given below:

Line 62: The authors describe opioid receptors MOR, DOR and KOR, but never mention the relative importance of the three receptors. Are there studies on different opioid receptor subtypes in neuropathic pain?

Line 87: The authors describe opioid receptor downregulation. The references are not discussed in detail. The following questions should be answered while citing an original paper: Which opioid receptors, following which conditions, in which animals, following how many days after injury etc.?

Line 110: The authors discuss epigenetic modifications focusing on DNMTs. DNA methyltransferases are a family of enzymes that catalyze the transfer of a methyl group to DNA. DNA methylation serves a wide variety of biological functions, not only limited to neuropathic pain. The focus of the review in this section switches to neuropathic pain but not on opioid receptors.

-The term “genetic level” is not appropriate. The authors should write “epigenetic modifications”.

-MiRNA activity is not a transcriptional but a post-translational modification.

Line 130: The authors again focus on neuropathic pain, not opioid receptors.

Line 183: The authors claim that opioid receptors in central nervous system are more likely to be downregulated after nerve injury. This needs to be explained in detail. Many of the important claims are not explained in detail throughout the manuscript.

-Line 58: What does the term “osteogenesis of NP” mean? Osteogenesis is the development and formation of bone.

-The conclusions are rather trivial and not helpful.

 -Two numbers are assigned to each reference – why?

-Table 1 is not helpful.  What do these genes have to do with opioid receptor downregulation?

In summary, the review should be rewritten with a focus on opioid receptors and opioids, in relation to neuropathic pain. More original research papers should be cited.

Author Response

Dear Editor and Reviewers,

Thanks very much for taking the time to review this manuscript. I appreciate all your valuable comments and suggestions! We have carefully considered these suggestions and make some supplements and changes to this manuscript as required. We have tried our best to improve and made some revisions.

Please find my itemized responses below (point-to-point) and my revisions/corrections in the re-submitted files.

Thanks again!

Best regards,

Lin Li

-------------------------

Reviewer 2 report:

The text is very difficult to follow. Extensive language corrections are necessary throughout the manuscript. The English needs to be checked by a native speaker.

We apologize for the poor language of our manuscript. We have modified the language of this manuscript to improve its readability.

The review is not focused on one specific topic. It rather is a combination of two different reviews: one on neuropathic pain and one on opioid receptors. Both topics are discussed poorly.

Your question is very good. Since we did not express it clearly, we are sorry for your misunderstanding. Our point of view is that the repeated use of opioids and NP will bring about a common pathological result, i.e., opioid receptor down-regulation. Therefore, we discussed the mechanisms of several opioid receptor down-regulation under these two conditions and cited relevant references. In order to help the readers to read and understand, we appropriately adjusted the order of the full text to make the expression more accurate and clearer, and hoping to increase the logic of the manuscript so that the readers can understand it clearly and with easy.

Large parts consist of unnecessary detailed information on well-known topics (e.g. receptor internalization), while others lack important information such as the mechanisms underlying opioid tolerance (e.g. line 218).

We are sorry for your misunderstanding. As repeated use of opioid drugs will promote the imbalance of the desensitization-internalization-resensitization cycle of opioid receptor, we think that the process of opioid receptor internalization is very important for maintaining the normal quantity and physiological function of membranous MOR. We also compared the differences between morphine and methadone in influencing MOR internalization and analgesia tolerance by citing literature. In addition, β-arrestin 2 plays an important role for internalization, and its role in opioid tolerance is discussed.

A connection between opioid receptors and neuropathic pain is described. However, endogenous opioids or opioid drugs are not discussed. Central or peripheral side effects of opioids (the main problem limiting opioid treatments in the clinical setting) are also not discussed in detail.

Thank you for your advice. We have added more details about the endogenous opioids and the side effects of opioids (Pages 2-3, lines 66-111).

A few examples for specific changes are given below:

Line 62: The authors describe opioid receptors MOR, DOR and KOR, but never mention the relative importance of the three receptors. Are there studies on different opioid receptor subtypes in neuropathic pain?

Thank you for pointing out our omissions, and we have added the content about opioid receptors in NP treatment in the manuscript (Page 3, line 78-111).

Line 87: The authors describe opioid receptor downregulation. The references are not discussed in detail. The following questions should be answered while citing an original paper: Which opioid receptors, following which conditions, in which animals, following how many days after injury etc.?

Thank you for your advice. We have added more details about the time-dependent decrease in opioid receptors in different pain models (Page 6, lines 199-208).

Line 110: The authors discuss epigenetic modifications focusing on DNMTs. DNA methyltransferases are a family of enzymes that catalyze the transfer of a methyl group to DNA. DNA methylation serves a wide variety of biological functions, not only limited to neuropathic pain. The focus of the review in this section switches to neuropathic pain but not on opioid receptors.

Line 130: The authors again focus on neuropathic pain, not opioid receptors.

Thank you for your good advice. We are sorry for your misunderstanding again. We have adjusted the order of the full text appropriately, focusing on the discussion of opioid receptors and/or factors directly affecting the expression of opioid receptors, hoping to increase the logicality of the article (Section 4, pages 6-8).

-The term “genetic level” is not appropriate. The authors should write “epigenetic modifications”.

This fault has been corrected (Page 7, line 226).

-MiRNA activity is not a transcriptional but a post-translational modification.

This fault has been corrected (Page 8, lines 283-288).

Line 183: The authors claim that opioid receptors in central nervous system are more likely to be downregulated after nerve injury. This needs to be explained in detail. Many of the important claims are not explained in detail throughout the manuscript.

Thank you for your advice. We have added more details about the opioid receptors in central nervous system in different pain models (Pages 8, lines 318-325). As the limited evidence about the down-regulation of opioid receptors in the central nervous system under neuropathic pain status, especially in the supraspinal regions, we can only describe it briefly here, and this will become the focus of future research.

-Line 58: What does the term “osteogenesis of NP” mean? Osteogenesis is the development and formation of bone.

Thank you very much for finding our mistakes in wording, which have now been corrected (Page 2, line 51).

-Two numbers are assigned to each reference – why?

Thank you very much for finding our mistakes in typesetting, which have now been corrected (Page 10, line 384).

-Table 1 is not helpful.  What do these genes have to do with opioid receptor downregulation?

Thank you for your advice. At present, the research on the mechanism of opioid receptor down-regulation in DRG involves different levels. This table is used to show the regulatory mechanism of different opioid receptors, so that readers can understand it more intuitively. In addition, we have also refined some contents in Table 1 (Page 6, line 217).

-The conclusions are rather trivial and not helpful.

In summary, the review should be rewritten with a focus on opioid receptors and opioids, in relation to neuropathic pain. More original research papers should be cited.

Thank you very much for your suggestions. We have made a lot of modifications for this review, added some related original articles and adjusted the order of the full text appropriately, focusing on the discussion of opioid receptors and opioids in the neuropathic pain status.

It seems that only one file can be uploaded here, so I can only upload the Marked-up version instead of the Clean copy. Please see the attachment.

Round 2

Reviewer 2 Report

The paper is still not focused on downregulation of opioid receptors during neuropathic pain following ongoing opioid treatment. The review should focus on this topic in the abstract and introduction, e.g., the mechanisms underlying opioid tolerance. There is abundant general information regarding neuropathic pain; however, opioid receptors, opioid drugs and their interaction is not discussed thoroughly. The conclusion is still not helpful. The authors should restructure the review and write in detail focusing on the above mentioned information. In its current form, the manuscript is not acceptable for publication.